# Recent Link Classification on Temporal Graphs Using Graph Profiler

## Abstract

The performance of Temporal Graph Learning (TGL) methods are typically evaluated on the *future link prediction* task, i.e., whether two nodes will get connected and *dynamic node classification* task, i.e., whether a node's class will change. Comparatively, *recent link classification*, i.e., to what class an emerging edge belongs to, is investigated much less even though it exists in many industrial settings. In this work, we first formalize recent link classification on temporal graphs as a benchmark downstream task and introduce corresponding benchmark datasets. Secondly, we evaluate the performance of state-of-the-art methods with a statistically meaningful metric *Matthews Correlation Coefficient*, which is more robust to imbalanced datasets, in addition to the commonly used average precision and area under the curve. We propose several design principles for tailoring models to specific requirements of the task and the dataset including modifications on message aggregation schema, readout layer and time encoding strategy which obtain significant improvement on benchmark datasets. Finally, we propose an architecture that we call *Graph Profiler*, which is capable of encoding previous events' class information on source and destination nodes. The experiments show that our proposed model achieves an improved Matthews Correlation Coefficient on most cases under interest. We believe the introduction of recent link classification as a benchmark task for temporal graph learning will be useful for the evaluation of prospective methods within the field.

## 1 Introduction

Graphs provide convenient structures to represent interactions or relationship between entities by modeling them as edges between vertices, respectively. Using this representation allows one to build models that capture the interconnected nature of complex systems such as social networks (El-Kishky et al., 2022; Wu et al., 2022; Gao et al., 2021) or transaction graphs (Liu et al., 2020; Zhang et al., 2022). Graph representation learning (GRL) rose in popularity due to the desire to apply deep learning to graph structured problems (Zhou et al., 2020; Wu et al., 2020; Hamilton, 2020). Indeed, GRL has provided significant advances in fraud detection (Liu et al., 2020; Zhang et al., 2022), recommendation systems (Wu et al., 2022; Gao et al., 2021), chemistry and materials science (Pezzicoli et al., 2022; Reiser et al., 2022; Bongini et al., 2021; Han et al., 2021; Xiong et al., 2021), traffic modeling (Rusek et al., 2019; Chen et al., 2022), and weather simulation (Keisler, 2022; Ma et al., 2022), among other possible applications. Many of these graph machine learning tasks can be understood as either link prediction (Chen et al., 2020; Cai et al., 2019; Zeb et al., 2022; Chamberlain et al., 2022) or node classification tasks (Kipf and Welling, 2016; Zhang et al., 2019), and make the assumption that the graph is static.

Acknowledging that many tasks in industrial settings involve graphs that evolve in time, researchers defined a sub-problem of graph machine called Temporal Graph Learning (TGL), with time dependent versions of the original static tasks, yielding future link prediction (FLP) and dynamic node classification (DNC) respectively (Kumar et al., 2019; Arnoux et al., 2017). The former task, FLP, seeks to predict whether two vertices will at some point be connected in the future; while the latter, DNC, seeks to predict whether vertices' class will change. Both of these tasks have a variety of applications in real world, e.g., predicting the probability of two people forming a tie or of a person deactivating their account on social media platforms, corresponding to FLP and DNC tasks respectively (Min et al., 2021; Song et al., 2021; Frasca et al., 2020; Zhang et al., 2021a).

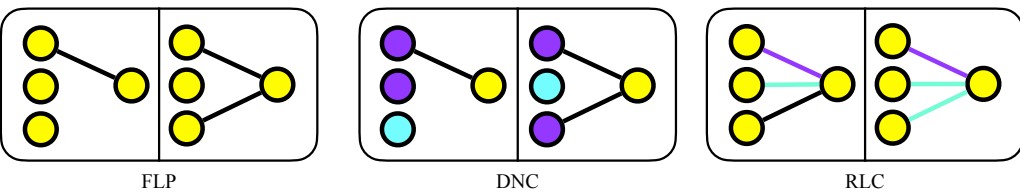

Figure 1: Differences between TGL tasks

This begs the question – "In analogy to the dynamic node classification task, is there a temporal link classification task we can define?" And indeed, there is a third common task that is based on static link classification (Wang et al., 2023), which we term recent link classification (RLC). RLC is present in industrial settings but has yet to be formalized in a research setting. The task of RLC requires that one classify a link that has been observed to exist. This task is important for settings where we wish to classify the interaction but labels may be delayed. In social media settings, this could involve classifying interactions as abusive, and in a transaction network it could involve classifying a transaction as potentially fraudulent.

More formally, we define RLC as a task in which the predictive algorithm is given the source and destination entities of a recent interaction and its features, e.g., textual content of the post; and it must predict the target class value. RLC tasks have typically been treated as tabular datasets focused on tasks such as fraud detection (Sarkar, 2022) or movie recommendations (Bennett et al., 2007; Harper and Konstan, 2015), where graph information has been largely ignored. For tabular tasks, it has been previously observed that counting features, such as how many times one user has previously liked another user's posts, provide significant metric uplift (Wu et al., 2019; Shan et al., 2016). While not explicitly graph machine learning, we take the view that these features are the result of manual feature engineering inspired by graph-based intuition such as the katz index (Zhang et al., 2021b; Martínez et al., 2016; Chamberlain et al., 2022). Therefore, we believe that bridging this gap and providing RLC tasks to the temporal graph machine learning community will push the community towards another industrially relevant problem.

With this motivation in hand, we formalize two research questions that we wish to answer in this work: "**Q1: How does recent link classification differ from future link prediction?**" and "**Q2: What are the most critical design principles of recent link classification?**". We answer the first question through a comparative study of baseline methods on both tasks; and we answer the second question by exploring a variety of modeling building blocks, some published previously, and some novel. In answering these research questions, we contribute a new temporal graph learning task, a new figure of merit, a measure of edge-homophily, and a non-exhaustive set of design principles that comprise a design space for this new machine learning task.

## 2  RELATED WORK

Our work should be understood within the context of the temporal graph learning literature. To that end, we have provided a brief review of the works that we have found influential. TGN (Rossi et al., 2020) is a message passing based encoder which learns graph node embeddings on a continuous-time dynamic multi-graph represented as a sequence of time-stamped events. It involves three main building blocks: (1) message function, (2) memory function and (3) embeddings module. At each event (e.g., a new node, node attribute change or new edge), a message function in the form of event-adopted MLPs, calculates aggregated messages to pass on parties involved (i.e., nodes). A memory function, such as an LSTM or GRU, updates the memory state of each node by the aggregated messages. An embedding module calculates the new state of node embeddings as a function of memory states and events. The authors experiment with simple embedding modules such as identity function which corresponds using memory states directly as well as multi-head attention based procedures. Their datasets include link prediction and node classification tasks. Gao and Ribeiro (2022) develops a framework to analyze the temporal graph learning architectures categorizing the methods literature into two groups 'time-and-graph' and 'time-then-graph'. Time-and-graph based architectures learns evolution of node representations by building a representation for each graph

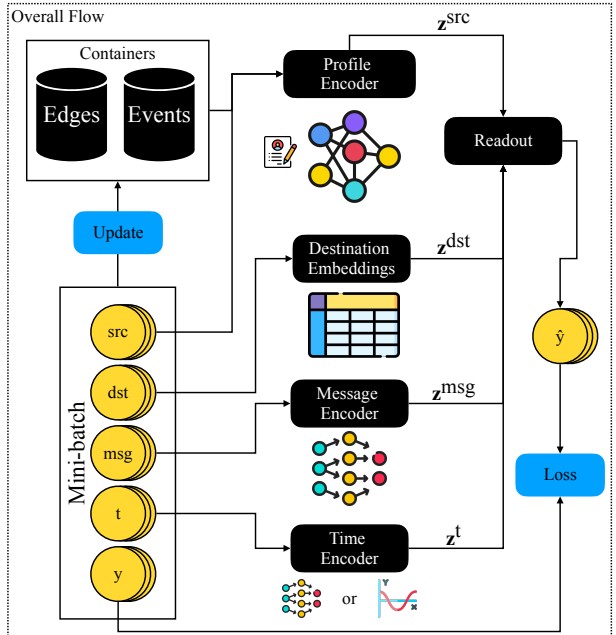
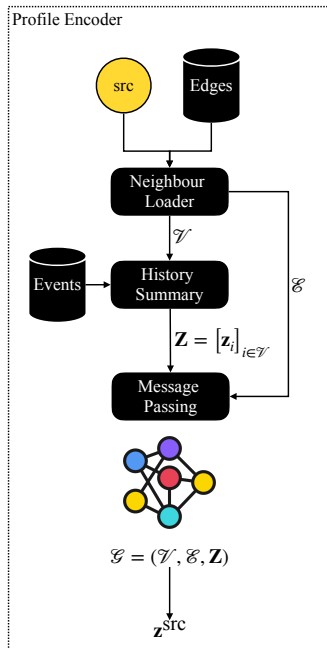

Figure 2: An overview of Profile Builder

snapshot. DySAT (Sankar et al., 2020) and EvolveGCN (Pareja et al., 2020) fall under this category. Time-then-graph based architectures, on the other hand, constructs a multi-graph by the memory of all past observations and build a static graph to learn the node representations on. TGAT (da Xu et al., 2020) and TGN (Rossi et al., 2020) fall under this category. In addition to the proposed framework that enables analyzing the expressivity of two distinct categories by the employed learnable modules, the authors suggest a simple but efficient algorithm GRU-GCN (Gao and Ribeiro, 2022). GraphMixer (Cong et al., 2023) takes a simpler view by constructing a model that has a fixed time encoding, alongside a node encoder and a link encoder, which are all fed into a link classifier that is trained to predict the existence of a link. Despite this simple infrastructure, GraphMixer is able to achieve state-of-the-art performance on both FLP and DNC tasks. Finally, TGL (Zhou et al., 2022) decomposes the task of training temporal graph neural networks into a general, scalable, framework composed of five core modules and provides primitives for computationally efficient training.

## 3 PROBLEM STATEMENT

A graph, $\mathcal{G}$, is composed of a set of vertices, $\mathcal{V}$, and edges $\mathcal{E}$, where each edge, $i \to j$, indicates a relationship between a pair of vertices $i$ and $j$. If a graph has an edge from $i$ to $j$ but no such edge in reverse, we say that the graph is directed. Each edge is assumed to be parameterized by a timestamp that indicates when that interaction occured. In most cases, the graph is constructed with entities as the vertices and interactions or relationships between those entities as edges. In the case of social networks vertices might be users and their posts, and edges might involve interactions either between users, such as follows or blocks, or interactions between users and posts such as likes or comments.

For a general RLC task we are given a set of source entities and destination entities, $\mathcal{V}_{\text{src}}$ and $\mathcal{V}_{\text{dst}}$, and a set of interactions between them $\mathcal{E} = \{(s_i, d_i, t_i, \mathbf{x}_i)\}_{i=1}^{M}$; such that interactions from source entity $s_i \in \mathcal{V}_{\text{src}}$ to destination entity $d_i \in \mathcal{V}_{\text{dst}}$ is realized at time $t_i$ and associated with raw feature vector of $\mathbf{x}_i \in \mathbb{R}^{d_{\text{msg}}}$ where $d_{\text{msg}}$ denotes the number. Given $m$ as the number of classes, each interaction is associated with a ground-truth target class represented as a binary vector $\mathbf{y}_i = (y_{i,1}, \ldots, y_{i,m})$ such that $y_{i,j} = 1$ if interaction $i$ belongs to $j^{\text{th}}$ class and $y_{i,j} = 0$ otherwise. The aim is learning a classifier that maps features, source and destination entities of an interaction to its target class value. Given a new interaction from source $s \in \mathcal{V}_{\text{src}}$ to destination $d \in \mathcal{V}_{\text{dst}}$ with features $\mathbf{x} \in \mathbb{R}^{d_0}$, let $\hat{\mathbf{y}} = (\hat{y}_1, \ldots, \hat{y}_m)$ denote the predicted target class likelihoods by the classifier, i.e., $f(\mathbf{x}, (s, d)) =$

$\hat{\mathbf{y}}$. Traditionally, the quality of estimation is evaluated by the cross entropy loss $\mathbb{L}_{\text{ce}}(\mathbf{y}, \hat{\mathbf{y}})$ during training:

$$\mathbb{L}_{\text{ce}}(\mathbf{y}, \hat{\mathbf{y}}) = -\sum_{j=1}^{m} y_j \log(\hat{y}_j). \tag{1}$$

While $G$ is not in general bipartite, we can choose to define the sets $\mathcal{V}_{\text{src}}$ and $\mathcal{V}_{\text{dst}}$ such that they overlap. This perspective provides us with the potential to learn different representations for a vertex as the sender and the reciever of a message. The source and destination entities do not hold features and the set of entities is static, therefore the problem is defined in transductive context. The raw edge features are observed at the time of event occurrence and do not change over time in addition to identity of source and destination entities, but the observation on edge labels are delayed.

## 4 GRAPH PROFILER

In this section, we introduce Graph Profiler, a simple architecture that is designed to learn entity profiles, or time-aggregated representations, over previous interactions and exploit that information to make classification decisions on the recent interactions along with the features of recent interaction. Graph Profiler is inspired by simple graph models that we have observed to work in webscale settings for industrial RLC tasks. Graph Profiler is composed of five main learnable modules; **profile encoder** $f_{\text{profile}}(\cdot)$, **message encoder** $f_{\text{msg}}(\cdot)$, **destination encoder** $f_{\text{dst}}(\cdot)$, **time encoder** $f_{\text{time}}(\cdot)$, and **readout** $f_{\text{rlc}}(\cdot)$, and two containers at time $t$; previous **events** $\mathbb{H}_t$ and meta-path **edges** $\mathbb{M}_t$.

For a given edge-set, $\mathcal{E}$, Graph Profiler proceeds in the following way to train. We begin by letting $\mathcal{E}_{\text{batch}} \subset \mathcal{E}$ denote a mini-batch of interactions with $t_{\text{current}} = \min_{j \in \mathcal{E}_{\text{batch}}} t_j$ denote the minimum interaction time in the batch, and $d_{\text{model}}$ denote the dimensionality of model. Given an interaction $(s_i, d_i, t_i, \mathbf{x}_i) \in \mathcal{E}_{\text{batch}}$, the computation proceeds through each module as:

1. The profile encoder calculates the source node profile $\mathbf{z}_i^{\text{src}} \in \mathbb{R}^{d_{\text{model}}}$ based on observations until $t_{\text{current}}$, i.e. $f_{\text{profile}}(s_i, \mathbb{H}_{t_{\text{current}}}) = \mathbf{z}_i^{\text{src}}$.

2. The message encoder encodes the interaction: $\mathbf{z}_i^{\text{msg}} \in \mathbb{R}^{d_{\text{model}}}$, i.e. $f_{\text{msg}}(\mathbf{x}_i) = \mathbf{z}_i^{\text{msg}}$.

3. The destination encoder generates up-to-date destination embeddings: $\mathbf{z}_i^{\text{dst}} \in \mathbb{R}^{d_{\text{model}}}$, i.e. $f_{\text{dst}}(d_i) = \mathbf{z}_i^{\text{dst}}$.

4. The time encoder converts the interaction timestamp into time embedding vector $\mathbf{z}_i^{\text{t}} \in \mathbb{R}^{d_{\text{model}}}$, i.e. $f_{\text{time}}(t_i) = \mathbf{z}_i^{\text{t}}$.

5. Readout layer to predict interaction class $\hat{\mathbf{y}}_i$ i.e. $f_{\text{rlc}}(\mathbf{z}_i^{\text{src}}, \mathbf{z}_i^{\text{msg}}, \mathbf{z}_i^{\text{dst}}, \mathbf{z}_i^{\text{t}},) = \hat{\mathbf{y}}_i$.

6. Once the predictions are made, the containers are updated with the mini-batch.

The overall flow is illustrated in Figure 2. Next we explain learnable module and the procedure to update containers.

**Profile Encoder**   Inspired by our previous experiences working on webscale recommendation systems, we derive graphs that allow us to capture source-source correlations that might be drowned out through traditional message passing schemes. Similar to previous work, we define a meta-path as an edge that is constructed from a path through the graph (Chen and Lei, 2022; Huai et al., 2023; Huang et al., 2022). In our specific instance, we consider second-order meta-paths that connect a vertex which acts as a source to another which acts as a course through a shared destination vertex. The set of meta-paths, $\mathbb{M}$, is time dependent because the edges are parameterized by time. Given the set of meta-path edges $\mathbb{M}_{t_{\text{current}}}$ observed until $t_{\text{current}}$, the profile encoder first queries the ego graph $\mathcal{G}_{t_{\text{current}}}(s_i)$ over the set of vertices $\mathcal{N}_{t_{\text{current}}}(s_i) = \{u : (u, s_i) \in \mathbb{M}_{t_{\text{current}}}\}$ and edge list $\mathcal{M}_{t_{\text{current}}}(s_i) = \{(u, v) : (u, v) \in \mathbb{M}_{t_{\text{current}}}\}$ for the source entity $s_i$. For each node $u \in \mathcal{N}_{t_{\text{current}}}(s_i)$, the set of relevant events are collected $\mathcal{H}_{t_{\text{current}}}(u) = \{(\mathbf{z}_i^{\text{dst}}, \mathbf{z}_i^{\text{msg}}, \mathbf{z}_i^{\text{t}}, \mathbf{y}_i) : (u, d_i) \in \mathbb{H}_{t_{\text{current}}}\}$. Thus, the ego graph of a given source node is composed of others that have interacted with the same destination nodes in the past and meta-paths within this neighbourhood. The node embeddings are initialized by aggregating the embeddings of previous events associated with the corresponding node, i.e. $\mathbf{h}^{(0)}(u) = f_{\text{aggregate}}(\mathcal{H}_{t_{\text{current}}}(u))$. For example, using a mean aggregation schema with single layer of linear transformation, the node embeddings $\mathbf{h}^{(0)}(u) \in \mathbb{R}^{d_{\text{model}}}$ are initialized as follows:

$$\mathbf{h}^0(u) = \frac{\sum_{i \in \mathcal{H}_{t_{\text{current}}}(u)} \left[\mathbf{z}_i^{\text{msg}} || \mathbf{z}_i^{\text{dst}} || \mathbf{z}_i^{\text{t}} || \mathbf{y}_i \right] \mathbf{W}_{\text{event}}^{\text{T}}}{||\mathcal{H}_{t_{\text{current}}}(u)||} \tag{2}$$

where $[\cdot || \cdot]$ denotes concatenation operator, $\mathbf{W}_{\text{event}} \in \mathbb{R}^{d_{\text{model}} \times d_1}$ are learnable weights with $d_1 = 3 \times d_{\text{model}} + m$, by concatenation e.g. $\mathbf{z}_i^{\text{msg}}, \mathbf{z}_i^{\text{dst}}, \mathbf{z}_i^{\text{t}} \in \mathbb{R}^{d_{\text{model}}}$ and $\mathbf{y}_i \in \mathbb{R}^m$. Then, using the GCN (Kipf and Welling, 2016) framework, at the $k^{\text{th}}$ layer the node embeddings are updated by passing messages between neighbour nodes, i.e. $\mathbf{h}^{(k)}(u) = f_{\text{gcn}}(\mathbf{h}^{(k-1)}(u), \mathcal{M}_{t_{\text{current}}}(s_i))$. For a normalized sum aggregation schema the update on node embeddings $\mathbf{h}^{(k)}(u) \in \mathbb{R}^{d_{\text{model}}}$ at layer $k$ looks as follows:

$$\mathbf{h}^{(k)}(u) = \sum_{(u,v) \in \mathcal{M}_{t_{\text{current}}}(s_i)} c_{u,v} \left(\mathbf{h}^{(k-1)}(v) \mathbf{W}_k^{\text{T}}\right) \tag{3}$$

such that $c_{u,v} = \frac{1}{\sqrt{\deg(u)} \cdot \sqrt{\deg(v)}}$ are normalization coefficients where $\deg(\cdot)$ denotes node degree on $\mathcal{G}$ and $\mathbf{W}_k \in \mathbb{R}^{d_{\text{model}} \times d_{\text{model}}}$ are learnable weights.

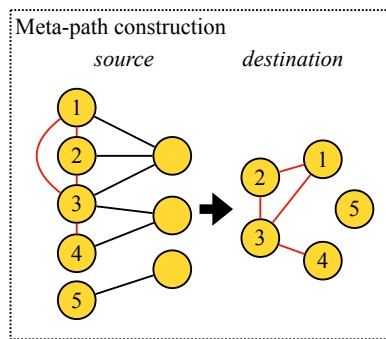

Figure 3: The source nodes that had interacted with same destination nodes are connected by meta-path edges.

**Other Modules**  The message encoder uses a single linear layer, such that; $f_{\text{msg}}(\mathbf{x}_i) = \mathbf{x}_i \mathbf{W}_{\text{msg}}^{\text{T}} + \mathbf{b}_{\text{msg}}$, where $\mathbf{W}_{\text{msg}} \in \mathbb{R}^{d_{\text{model}} \times d_{\text{msg}}}$ and $\mathbf{b}_{\text{msg}} \in \mathbb{R}^{d_{\text{model}}}$ are learnable weights. For destination encoding, we use an embeddings look-up table of size equal to number of destination nodes and initialized randomly, such that; $f_{\text{dst}}(d_i) = \mathbb{1}_{d_i} \mathbf{W}_{\text{dst}}^{\text{T}}$, where $\mathbb{1}_{d_i} \in \mathbb{R}^{||\mathcal{V}_{\text{dst}}||}$ denotes one-hot vector representation of node $d$, and $\mathbf{W}_{\text{dst}} \in \mathbb{R}^{d_{\text{model}} \times ||\mathcal{V}_{\text{dst}}||}$ denotes learnable weights. For the time encoder, we employ either fixed time encoding function proposed by Cong et al. (2023) or learnable time projection introduced by Kumar et al. (2019). The time encoding formulations are provided in Appendix B. The predictions at readout layer are simply made as $\hat{\mathbf{y}}_i = \left[\mathbf{z}_i^{\text{src}} + \mathbf{z}_i^{\text{dst}} + \mathbf{z}_i^{\text{msg}} + \mathbf{z}_i^{\text{t}}\right] \mathbf{W}_{\text{rlc}}^{\text{T}}$ where $\mathbf{W}_{\text{rlc}} \in \mathbb{R}^{d_{\text{model}} \times d_{\text{model}}}$.

**Container Updates**  After performing backpropagation over a mini-batch, it is inserted into the previous events container $\mathbb{H} := \mathbb{H} \cup \mathcal{E}_{\text{batch}}$. The edge container is also updated by re-calculating the meta-paths given new observations. Presence of a meta-path edge between two source entities refer to the event that they have interacted with the same destination entity in the past (see Figure 3).

Unlike static graph learning methods, Graph Profiler is capable of encoding temporal properties of network if useful for making edge classification decisions. Graph Profiler has two main advantages over using existing temporal graph learning methods: (1) it enables building dynamic entity profiles over previous interactions' feature and label information of their neighbourhoods and (2) is capable of maintaining a long-term view of an entity's profile that could capture long-term preferences. In addition, the modular structure of Graph Profiler is flexible to adapt the model to contextual properties of dataset with suggested variants.

## 5  EXPERIMENTS

In order to understand RLC as a novel task within temporal graph learning, we begin by evaluating a two layer MLP, TGN (Rossi et al., 2020), TGAT (da Xu et al., 2020), and Graph Mixer (Cong et al., 2023) on RLC by making the appropriate modifications. We have chosen these methods because they are state-of-the-art TGL baselines developed for FLP. With these results in hand, we outline a set of design principles that comprise the design space for RLC. With these design principles in mind, we present Graph Profiler and benchmark it on six different datasets. For each dataset, we locate the optimal portion of our design space and discuss the correspondance between that and the underlying dynamics of the dataset under investigation.

Table 1: Dataset statistics

|  | $|\mathcal{V}_{\mathrm{src}}|$ | $|\mathcal{V}_{\mathrm{dst}}|$ | $|\mathcal{E}|$ | $d_{\mathrm{msg}}$ | $\rho$ | $\bar{\mathcal{H}}_e$ | $\mathcal{H}_e^+$ | $\mathcal{H}_e^-$ | $\tilde{\mathcal{H}}_e$ |
|---|---|---|---|---|---|---|---|---|---|
| EPIC GAMES | 542 | 614 | 17584 | 400 | 0.6601 | 0.8330 | 0.9038 | 0.6955 | 0.7663 |
| YELPCHI | 38,063 | 201 | 67,395 | 25 | 0.1323 | 0.7781 | 0.1589 | 0.8725 | 0.2533 |
| WIKIPEDIA | 8,227 | 1,000 | 157,474 | 172 | 0.0014 | 0.9975 | 0.0130 | 0.9988 | 0.0144 |
| MOOC | 7,047 | 97 | 411,749 | 4 | 0.0099 | 0.9809 | 0.0212 | 0.9904 | 0.0308 |
| REDDIT | 10,000 | 984 | 672,447 | 172 | 0.0005 | 0.9989 | 0.0025 | 0.9995 | 0.0030 |
| OPEN SEA | 57,230 | 1,000 | 282,743 | 35 | 0.4601 | 0.5865 | 0.5505 | 0.6171 | 0.5812 |

$|\mathcal{V}_{\mathrm{src}}|$ denotes number of source entities, $|\mathcal{V}_{\mathrm{dst}}|$ denotes number of destination entities, $d_{\mathrm{msg}}$ denotes number of interaction features, $|\mathcal{E}|$ denotes number interactions, $\rho$ denotes the ratio of positive class, $\bar{\mathcal{H}}_e$, $\mathcal{H}_e^+$, $\mathcal{H}_e^-$, and $\tilde{\mathcal{H}}_e$ denotes the average, positive, negative and balanced edge homophily levels at the final graph, respectively.

**Datasets** We evaluated our methods on four benchmark datasets that have previously been used by the TGL community – YELPCHI (Dou et al., 2020), WIKIPEDIA, MOOC, and REDDIT (Kumar et al., 2019). We convert these FLP datasets into RLC datasets on a case-by-case basis. On YELPCHI, the source entities are platform users and destination entities include hotels and restaurants. An interaction happens when a user reviews one of the hotels or restaurants. The reviews are labeled either as filtered (spam) or recommended (legitimate). For WIKIPEDIA, the set of entities is composed of users and pages, and an interaction happens when a user edits a page. For REDDIT, entities are users and subreddits, and an interaction represents a post written by a user on a subreddit. Some page edits on Wikipedia and posts on Reddit may be controversial causing the user to be banned. Thus, on both datasets we base the target class of interaction on whether it is controversial or not, i.e., whether it got the user banned. The interaction features on these three datasets are extracted based on the textual content of edit/post/review. The MOOC dataset consists of actions done by students on a MOOC online course. The source entities are defined by students and the destination entities are defined by course contents the students interact with, such as; recordings, lecture notes etc. The interaction features are defined by the type of activities the student perform on an interaction, e.g., viewing the video, submitting an answer on the forum etc. Sometimes, the students dropout the course after an activity, which we use to identify the target class of interaction. Thus, all four datasets are binary recent link classification datasets on which the class imbalance is salient.

In addition to adapting benchmark TGL datasets to RLC task, we process two tabular datasets that are not conventionally investigated in TGL setting; EPIC GAMES [1] and OPEN SEA (La Cava et al., 2023a;b; Costa et al., 2023) [2] The Epic Games Store is a digital video game storefront, operated by Epic Games. The dataset includes the critics written by different authors on the games released on the platform. The source and destination nodes represent authors and games respectively and critics form the set of interactions. We vectorize the textual content of critics using TF-IDF features to use as interaction features and involved the overall rating the author provided as well. The label of the interaction is determined based on whether it was selected as top critic or not. Given the fact that once a critic is released all the information regarding the author, game and features of the critic is available, but whether it will end up being selected as top-critic is a delayed observation, the data naturally forms into an RLC task. Open Sea is one of the leading trading platform in the Web3 ecosystem, a collection of Non-Fungible Tokens (NFT) transactions between 2021 and 2023 sourced from Open Sea is provided as natural language processing dataset and is mainly being used for multimodal learning classification tasks. Best to our knowledge the dataset is not investigated in TGL framework so far. In order to do so we have identified disjoint sets of sellers and buyers of unique NFTs (that are identified by collection memberships and token IDs) to serve as source and destination nodes. The transaction features are calculated as binary representation of categorical variable fields on the data and cryptocurrency exchange rates at the time of interaction in addition to monetary values associated such as seller price and platform fees. The labels of transactions are determined based on the future transaction of unique NFTs such that it is tagged as 'profitable' if the revenue obtained at the sale was higher than the price paid at the purchase. Therefore, the very last transaction of each unique NFT is discarded. Again, as the labels are delayed, i.e., whether it will be a profitable investment is not know at the time of purchase, in contrast to transaction features, the

---

[1]https://www.kaggle.com/datasets/mexwell/epic-games-store-dataset.
[2]https://huggingface.co/datasets/MLNTeam-Unical/NFT-70M_transactions.

data fits well to RLC setting. The link to processed versions of these datasets are available at the repository of this work.

The datasets statistics are provided in Table 1. In our experiments, data is divided into training (70%), validation (10%) and testing (20%) sets chronologically.

**Edge Homophily**  We introduce a measure of edge homophily to understand the importance of graph information to our edge classification task. Our edge homophily measure is defined as:

$$\bar{\mathcal{H}}_e(\mathcal{G}) = \frac{1}{|\mathcal{E}|} \sum_{\alpha \in \mathcal{E}} \sum_{\beta \in \mathcal{N}_\alpha^{(e)}} \frac{\mathbf{1}_{l(\alpha)=l(\beta)}}{|\mathcal{N}_\alpha^{(e)}|} \tag{4}$$

where $\mathcal{N}^e$ is the edge-wise neighbourhood operator and $l$ is the operator that returns the label of the edge. The edge-wise neighbourhood operator constructs a set of all edges that are connected to a given edge, $\alpha = (i,j)$, where $i$ and $j$ are the source and destination respectively, by constructing the union of two sets $\mathcal{N}^e(\alpha) = \mathcal{I}(i) \cup \mathcal{O}(j)$, where $\mathcal{I}(\cdot)$ and $\mathcal{O}(\cdot)$ construct the set of incoming and outgoing edges respectively. For the simplicity of notation, we have neglected the time dimension but this definition is easy to generalize to temporal graphs through the neighbourhood operators.

Edge-homophily measures the fraction of edges that connect nodes of the same class in analogy to node-homophily measurement which is a critical dataset property that determines the importance of encoding graph structure in node classification tasks (Pei et al., 2020). The edge homophily definition in equation 4 treats different classes equally, which may be misleading for unbalanced datasets. Therefore, we also define balanced edge homophily for binary classification as $\tilde{\mathcal{H}}_e(\mathcal{G}) = (1 - \rho)\mathcal{H}_e^+ + \rho\mathcal{H}_e^-$ where $\rho$ denotes the ratio of positive class and $\mathcal{H}_e^+$, ($\mathcal{H}_e^-$) positive (negative) class edge homophily levels. We calculate the edge homophily levels presented in Table 1 by the final graph. In Appendix A, we also provide the dynamics of edge homophily over time.

**Performance Evaluation**  Two of the most common performance metrics used for performance evaluation for both FLP and DNC are area under the receiver operating characteristic curve (AUC) and average precision score (APS), but these metrics are well known to saturate in the case of AUC or provide skewed measures of quality in the case of imbalanced datasets (Chicco and Jurman, 2020; 2023). As a result, we turn to the Matthews Correlation Coefficient (Yule, 1912; Gorodkin, 2004), which is defined as:

$$\text{MCC} = \frac{cs - \vec{t} \cdot \vec{p}}{\sqrt{s^2 - \vec{p} \cdot \vec{p}}\sqrt{s^2 - \vec{t} \cdot \vec{t}}}$$

where $\vec{t}$ is a vector of the number of times each class occurred, $\vec{p}$ is a vector of the number of times each class was predicted, $c$ is the number of samples correctly predicted, and $s$ is the total number of samples. This correlation coefficient always has a maximum value of 1, and the minimum value ranges between -1 and 0, depending on the distribution of the underlying data. A score of 0 indicates that the predictor is perfectly random; a score of 1 indicates that the predictor is perfectly accurate; and a score of -1 indicates that the predictor is perfectly inaccurate. As an illustrative example, we present use case A1 from Table 4 in Chicco and Jurman (2020). In this example, we have 100 total data points with 91 in the positive class and 9 in the negative. For a hypothetical classifier that that predicts all but one data point as a member of the positive class; we find $TP = 90, FN = 1, TN = 0, FP = 9$. This yields a respectable APS of 0.90 but a near random MCC of -0.03. While simple, this is just one example where metrics like APS can mask underlying poor performance for imbalanced datasets. Chicco and Jurman (2023) further presents similar failure modes for ROC-AUC. Because of this, we choose MCC as our figure of merit for the RLC tasks that we present.

**Implementation**  In an effort to present fair comparisons, we performed 100 steps of hyperparameters optimization to optimize the hyperparameters of all models using the software package OPTUNA (Akiba et al., 2019). Each experiment was run over the same 10 seeds. All tuning was performed on the validation set where we maximize the average accuracy across all 10 seeds, and we report the test-seed averaged results on the test set that are associated with those hyperparameter settings that maximize the validation accuracy. All models were implemented using PYTORCH GEOMETRIC 2.3.1 (Fey and Lenssen, 2019) and PYTORCH 1.13 (Paszke et al., 2019). We imple-

Table 2: TGN Variants

|  |  | YELPCHI | | | WIKIPEDIA | | | MOOC | | | REDDIT | | |
|---|---|---|---|---|---|---|---|---|---|---|---|---|---|
|  |  | MCC | APS | AUC | MCC | APS | AUC | MCC | APS | AUC | MCC | APS | AUC |
| Time Encoding | fix | 0.2624 | 0.3148 | 0.7590 | **0.2943** | **0.1237** | **0.9086** | **0.1004** | 0.0486 | 0.7634 | 0.0042 | 0.0049 | **0.6608** |
|  | learn | **0.2866** | **0.3278** | **0.7723** | 0.1933 | 0.0989 | 0.8728 | 0.0973 | **0.0571** | **0.7730** | **0.0444** | **0.0093** | 0.6508 |
| Aggregator | exp | 0.2803 | 0.3262 | 0.7700 | 0.1018 | 0.0712 | 0.8653 | **0.0630** | **0.0415** | **0.7494** | 0.0158 | 0.0036 | **0.6608** |
|  | last | **0.2866** | **0.3278** | **0.7723** | **0.2943** | **0.1237** | **0.9086** | 0.0477 | 0.0325 | 0.7045 | **0.0444** | 0.0055 | **0.6599** |
|  | mean | 0.2744 | 0.3217 | 0.7666 | 0.2034 | 0.0896 | 0.8955 | 0.1004 | 0.0571 | 0.7730 | 0.0142 | **0.0093** | 0.6235 |
| Readout | src | 0.2286 | 0.2391 | 0.7096 | 0.1237 | 0.0828 | 0.8368 | 0.0530 | 0.0416 | 0.7199 | 0.0105 | 0.0045 | 0.6435 |
|  | dst | 0.2288 | 0.2311 | 0.7015 | 0.0972 | 0.0464 | 0.7298 | 0.0432 | 0.0377 | 0.7195 | 0.0099 | 0.0049 | 0.6188 |
|  | src-dst | 0.2319 | 0.2411 | 0.7094 | 0.1018 | 0.0355 | 0.8908 | 0.0924 | 0.0462 | 0.7485 | 0.0142 | 0.0031 | 0.6608 |
|  | src-t | 0.2311 | 0.2426 | 0.7147 | 0.1308 | 0.0844 | 0.8401 | 0.0507 | 0.0386 | 0.7191 | 0.0104 | 0.0021 | 0.6472 |
|  | dst-t | 0.2277 | 0.2381 | 0.7063 | 0.1057 | 0.0469 | 0.7379 | 0.0338 | 0.0411 | 0.7205 | 0.0159 | 0.0124 | 0.6211 |
|  | src-dst-t | 0.2290 | 0.2542 | 0.7151 | 0.1442 | 0.0346 | 0.9086 | 0.0903 | 0.0506 | 0.7729 | 0.0444 | 0.0092 | 0.6508 |
|  | src-msg | 0.2732 | 0.3166 | 0.7627 | 0.1530 | 0.0835 | 0.8808 | 0.0996 | 0.0603 | 0.7763 | 0.0074 | 0.0024 | 0.6046 |
|  | dst-msg | 0.2744 | 0.3209 | 0.7641 | 0.1184 | 0.0564 | 0.8332 | 0.0475 | 0.0289 | 0.7112 | 0.0171 | 0.0144 | 0.5987 |
|  | src-dst-msg | 0.2644 | 0.3147 | 0.7629 | 0.2040 | 0.0714 | 0.8537 | 0.0894 | 0.0497 | 0.7708 | 0.0158 | 0.0033 | 0.6599 |
|  | src-msg-t | 0.2866 | 0.3278 | 0.7723 | 0.1764 | 0.0858 | 0.8994 | 0.1004 | 0.0571 | 0.7727 | 0.0040 | 0.0014 | 0.6005 |
|  | dst-msg-t | 0.2803 | 0.3230 | 0.7669 | 0.1300 | 0.0617 | 0.7489 | 0.0462 | 0.0325 | 0.7045 | 0.0113 | 0.0093 | 0.5971 |
|  | src-dst-msg-t | 0.2734 | 0.3217 | 0.7666 | 0.2943 | 0.1237 | 0.9020 | 0.0973 | 0.0536 | 0.7730 | 0.0077 | 0.0049 | 0.6089 |

mented TGN using the layers that are publicly available in PYTORCH GEOMETRIC, GraphMixer [3] and TGAT [4] were implemented using the authors opensource implementation provided on their github repository. All computations were run on an `Nvidia DGX A100` machine with 128 `AMD Rome 7742` cores and 8 `Nvidia A100` GPUs.

## 5.1 KEY FACTORS TO TAILOR MODEL TO SPECIFIC NEEDS OF DATA

With the introduction of RLC as a benchmark task to evaluate temporal graph learning methods, we explore the performance of a well-known state-of-the art model, TGN (Rossi et al., 2020)and variations. We create these variants by constructing different message aggregation schema, readout layers, and time encoding strategy to better discover the potential of overall architecture. More specifically, we experiment with (1) fixed and learnable time encoding as proposed by Cong et al. (2023) and by Kumar et al. (2019), respectively. We use mean, last and exponential decay message aggregators. The mean and last aggregators calculate state of a node by the average and by the last of interactions in memory, respectively, as described by Rossi et al. (2020). Exponential decay variant calculates a weighted average over interactions in the memory by setting weights so that they decrease exponentially by their associated timestamp. Finally, we try six different configurations on the inputs of readout layer based on the combinations of source, destination, time and message embeddings calculated for the recent event. In Table 2, the results are summarized and in Appendix E violin plots for each variation under interest are provided.

The readout variations appeared to play a significant role in the model's performance as can be seen in Figure 6. In Figure 6, the blue glyphs correspond to combinations of the vertex features, while the red glyphs correspond to combinations of the vertex *and* message features. The star, circle, and triangle glyphs correspond to the src-dst, src, and dst embeddings respectively. These results on the WIKIPEDIA dataset are demonstrated on different levels of model dimension, i.e. $d \in \{100, 200\}$ by various levels of number of neighbours. We observe that incorporating the edge features as residual at the final layer of update looks helpful for improving the performance in terms of MCC, which makes intuitive sense given that the message features for this dataset correspond to the edit's content. Interestingly, we only observe this trend when looking at the MCC curves where we see a dramatic stratification in performance. We see similar trends in the APS and Loss curves (See Appendix F). The AUC curves show an opposite trend, which we interpret as AUC being an unsatisfactory metric for the evaluation of RLC tasks.

We conclude that on abuse-related datasets the most recent interaction matters most therefore aggregation based on last event is more useful. In the case of predicting course completion, the average of previous actions are valuable, which is captured by the importance of the mean aggregator. In general, we observe that involving recent interaction features in readout layer is very useful as the configurations with `msg` perform significantly better.

---

[3] `https://github.com/CongWeilin/GraphMixer`
[4] `https://github.com/StatsDLMathsRecomSys/Inductive-representation-learning-on-temporal-graph`

Table 3: Model Comparison

|     |            | EPIC GAMES | YELPCHI | WIKIPEDIA | MOOC | REDDIT | OPEN SEA |
|-----|------------|-----------|---------|-----------|--------|--------|----------|
| MCC | MLP | 0.1554 | 0.2763 | 0.2354 | 0.0673 | 0.0021 | 0.1106 |
|     | TGN | 0.8373 | 0.2372 | 0.1442 | 0.0991 | 0.0174 | 0.1071 |
|     | TGAT | 0.5546 | 0.1890 | 0.0000 | 0.0000 | 0.0000 | 0.2245 |
|     | Graph Mixer | 0.2316 | 0.2830 | 0.1442 | 0.1174 | 0.0000 | 0.2647 |
|     | Modified TGN | 0.8713 | 0.2866 | 0.2943 | 0.1004 | 0.0444 | 0.2647 |
|     | Graph Profiler | 0.9355 | 0.3274 | 0.2498 | 0.1739 | 0.0115 | 0.2959 |
| APS | MLP | 0.6976 | 0.3254 | 0.0759 | 0.0169 | 0.0012 | 0.7790 |
|     | TGN | 0.9850 | 0.2461 | 0.0320 | 0.0526 | 0.0018 | 0.7912 |
|     | TGAT | 0.8844 | 0.1789 | 0.0157 | 0.0190 | 0.0014 | 0.6582 |
|     | Graph Mixer | 0.7522 | 0.3252 | 0.0550 | 0.0711 | 0.0021 | 0.8304 |
|     | Modified TGN | 0.9892 | 0.3249 | 0.1148 | 0.0430 | 0.0055 | 0.8225 |
|     | Graph Profiler | 0.9988 | 0.4059 | 0.0955 | 0.0896 | 0.0092 | 0.8459 |
| AUC | MLP | 0.6117 | 0.7694 | 0.7731 | 0.6447 | 0.5486 | 0.5776 |
|     | TGN | 0.9734 | 0.7135 | 0.7135 | 0.7672 | 0.5671 | 0.5908 |
|     | TGAT | 0.8470 | 0.6314 | 0.8908 | 0.6383 | 0.5336 | 0.6871 |
|     | Graph Mixer | 0.7132 | 0.7650 | 0.7500 | 0.7515 | 0.6413 | 0.6789 |
|     | Modified TGN | 0.9807 | 0.7723 | 0.7723 | 0.7439 | 0.6508 | 0.6544 |
|     | Graph Profiler | 0.9974 | 0.8058 | 0.7821 | 0.7886 | 0.6280 | 0.6740 |

For each dataset and metric, the best results are coloured red and second best results are coloured blue.

## 5.2 MODEL COMPARISON

Using the results from the TGN modifications, we have identified multiple core design principles associated with improved performance on the RLC task, and we have incorporated these principles into Graph Mixer. Specifically, we observed that generally a learnable time-encoding provided improved results for three of four data sets; and the `src-dst-msg-t` read-out variant provided generally strong results across all four data sets. Because of these results, we designed Graph Profiler with a learnable time encoder and a `src-dst-msg-t` readout function. To validate these results, we perform benchmark experiments of baselines and Graph Profiler introduced in section 4 on RLC. Between the existing benchmark datasets, on YELPCHI and MOOC, we observe that the Graph Profiler obtains the best results while on WIKIPEDIA and REDDIT, our modifications to the TGN architecture are more useful. The most probable reason is that graph bipartiteness (i.e., ratio of number of source nodes to number of destination nodes) is much higher on YELPCHI and MOOC compared to WIKIPEDIA and REDDIT, and Graph Profiler is designed to operate on the bipartiteness of interaction graphs. Therefore, the destination node embeddings are feasible to look up. Another observation we draw from these results of model comparison is the usefulness of MCC to demonstrate the discrepancy of different models. For instance, on MOOC Graph Profiler improves MCC by 73% on top of Modified TGN, while the change on AUC is only by 3% (See Table 3). On EPIC GAMES and OPEN SEA datasets which are less unbalanced compared to others, Graph Profiler outperforms other baselines consistently based on MCC and APS, while the ranking deviates based on AUC in small margins. Revisiting the dataset statistics provided in Table 1, we conclude that encoding node profiles based on their shared history, i.e., employing Graph Profiler architecture, is more effective on datasets with higher balanced edge homophily and less class imbalance compared to tracking a graph-wide memory, i.e., employing TGN framework.

## 6 CONCLUSION

In this work, we introduce recent link classification on temporal graphs as a benchmark downstream task and evaluate the most competitive state-of-the-art method's performance using a statistically meaningful metric *Matthews Correlation Coefficient*. This metric is more robust to imbalanced datasets, in comparison to the commonly used average precision and area under the curve. We propose several design principles for tailoring models to specific requirements of the task and the dataset based on message aggregation schema, readout layer and time encoding strategy which obtain significant improvement on benchmark datasets. We present an architecture that we call *Graph Profiler* a recent link classification algorithm designed for bipartite graphs which is common in industrial settings. We believe the introduction of recent link classification as a benchmark task for temporal graph learning is useful for the evaluation of prospective methods within the field.

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

# APPENDICES

## A   EDGE HOMOPHILY TRENDS IN DATASETS

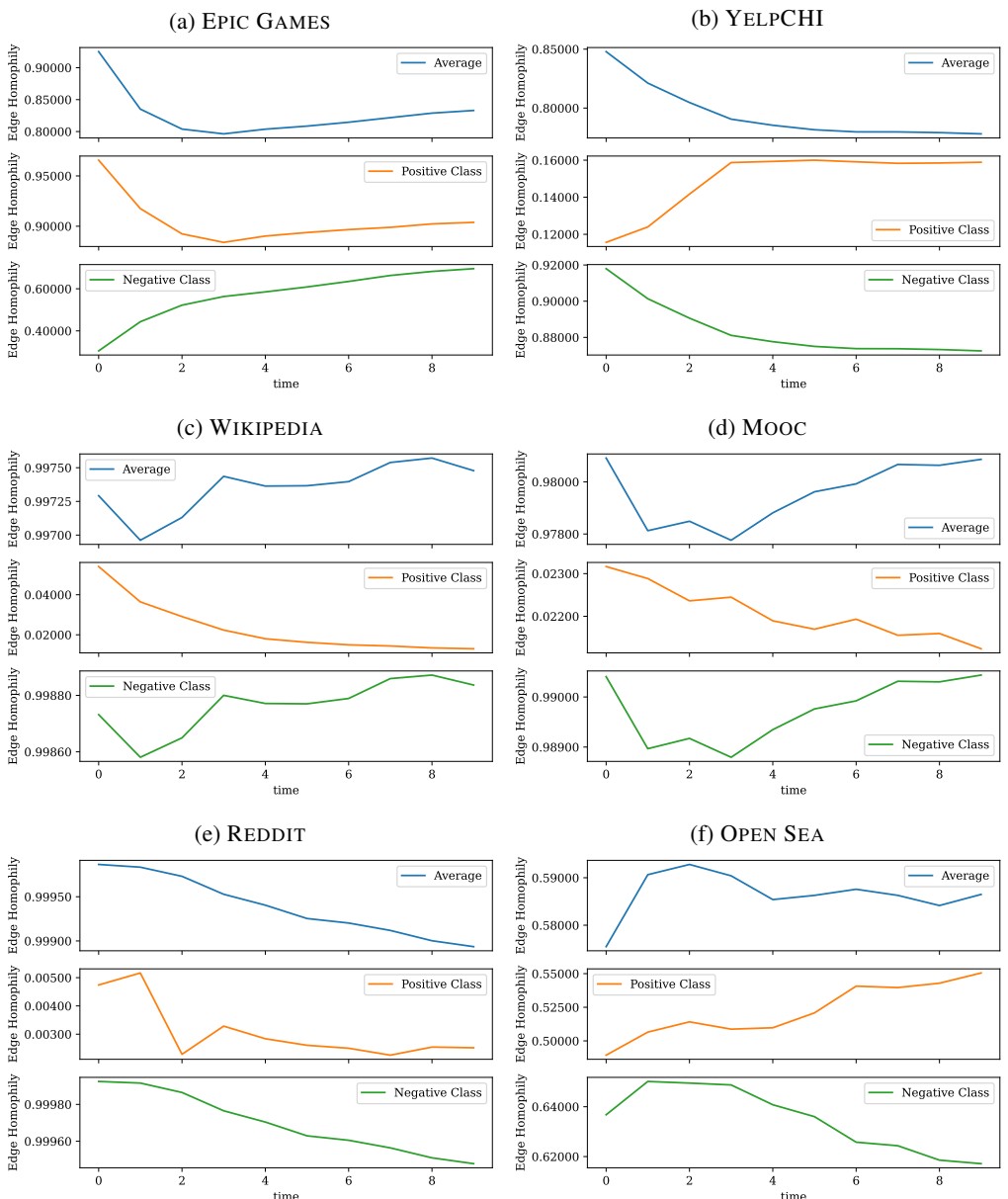

Figure 4: Edge Homophily Trends

## B  TIME ENCODING FORMULATIONS

Given a weight vector $\omega \in \mathbb{R}^{d_{\text{model}}}$, in general the time encoding function follows:

$$f_{\text{time}}(t) = \cos(t\omega) = \mathbf{z}^t$$

where $\mathbf{z}^t \in \mathbb{R}^{d_{\text{model}}}$ denotes the vector representation of timestamp $t$. The two variants of time encoding we investigate in the scope of this work differs in calculation of weight vector. In the *learnable version*, $\omega \in \mathbb{R}^{d_{\text{model}}}$ is simply learned during training, so the time encoding layer is a linear projection without the bias parameter followed with cosine scaling. In the case of *fixed time encoding* as proposed by Cong et al. (2023), each dimension of weight vector is given the feature $\omega_i = \alpha - \frac{(i-1)}{\beta}$ so that $t\omega$ is a vector with monotonically exponentially decreasing values. The $\alpha$ and $\beta$ are hyperparameters to be selected depending on the scale of the minimum and maximum timestamps in the data. In practice $\alpha = \beta = \sqrt{d_{\text{model}}}$ is found to perform well by the authors, which we also follow in our experiments.

## C  ABLATION STUDY

In order to investigate the impact of learnable destination embeddings, we experiment with and without learning a set of embeddings for the destination nodes. The results are presented in Table 4. For most of the datasets, learnable destination node representations which are then used for building source node profiles improves the predictive performance of Graph Profiler, with an exception on REDDIT dataset. It can be inferred that destination encoding enriches the source profile embeddings by temporally smoothing the interactions to build a sense of history, by contrast the destination in the readout itself is only a point in time estimate. We believe that the difference on REDDIT dataset may be relevant with the fact that whether a sub-reddit is controversial is less dependent on the main post compared to Wikipedia case. Said another way, there are many wikipedia pages that are prone to abuse for political reasons, and thus, the page profile matters. By contrast, abuse on reddit is less dependent on the subreddit than the author.

Table 4: The impact of using destination embeddings

|  | MCC | | APS | | AUC | |
|---|---|---|---|---|---|---|
|  | *with* | *without* | *with* | *without* | *with* | *without* |
| EPIC GAMES | **0.9355** | 0.7695 | **0.9988** | 0.9575 | **0.9974** | 0.9054 |
| YELPCHI | **0.3274** | 0.3071 | **0.4059** | 0.3892 | **0.8058** | 0.8000 |
| WIKIPEDIA | **0.2498** | 0.1324 | **0.0955** | 0.0366 | **0.7821** | 0.6946 |
| MOOC | **0.1739** | 0.0000 | **0.0896** | 0.0011 | **0.7886** | 0.5906 |
| REDDIT | 0.0115 | **0.0701** | 0.0092 | **0.0203** | 0.6280 | **0.6833** |

## D ON THE IMPORTANCE OF HYPERPARAMETER SENSITIVITY DIFFERENCES BETWEEN FLP AND RLC

In our reproduction study we explored the effect of negative sampling ratio, batch size, and the number of sampled neighbors on the performance of our TGN baseline for the FLP task. We term these as non-architectural parameters because they influence the training but do not influence the architecture of the model itself. We decided to explore these parameters because they represent a tradeoff between computational performance and utilization; and model accuracy. In the example of batch size, this is typically tuned to be as large as possible to maximize GPU utilization but we see in Figure 5, observe a steady decline in MCC (other metrics can be found in the appendix) as the batch size is increased. Indeed, we observe variation in model performance due to changes in batch size that are larger than the variations that come from new model architectures. Intuitively the decay makes sense, because the gradient updates become less frequent, but points to a relatively significant but under-discussed tradeoff that has real ramifications for production use-cases. In the case of negative sampling ratio, we observe a slight decline in MCC and a decrease in the consistency between individual training runs as the number of negative samples increases. Thus, it can be concluded that RLC does not have the same dependence on batch-size and does not require the generation of negative samples. These lead us to the conclusion that the assumptions made during the development of models for FLP may not hold for RLC, and direct translation of existing TGL methods, which are generally benchmarked on FLP tasks, to perform RLC in industrial setting is not convenient. Because of this, we believe that RLC is an interesting general purpose benchmark task for TGL community to treat differently from the commonly used ones. The results on other datasets are provided in Appendix G.

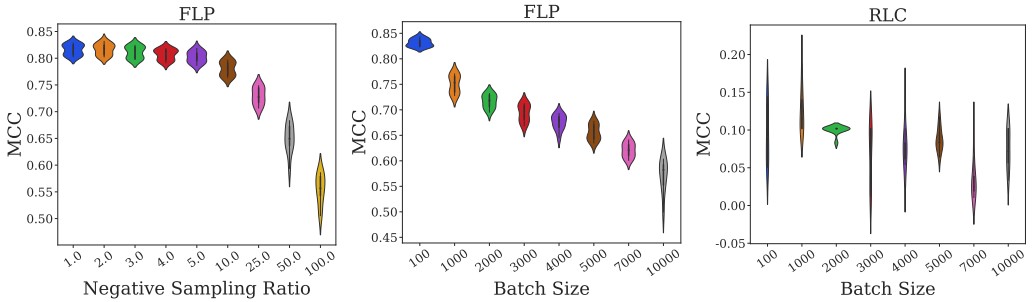

Figure 5: TGN performance on FLP task in terms of MCC metric on Wikipedia dataset with varying levels of negative sampling ratio, batch size, and number of neighbors over 10 different random seed initialization.

## E TGN MODIFICATIONS ON RLC

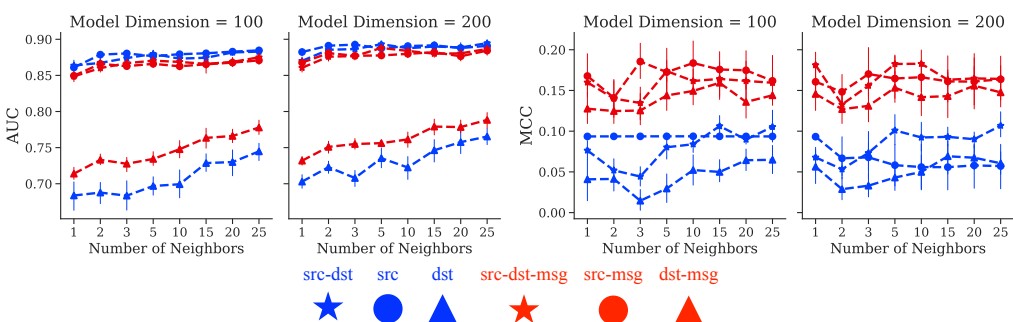

Figure 6: Readout variations on Wikipedia. The blue glyphs correspond to combinations of the vertex features, while the red glyphs correspond to combinations of the vertex *and* message features. The star, circle, and triangle glyphs correspond to the src-dst, src, and dst embeddings respectively.

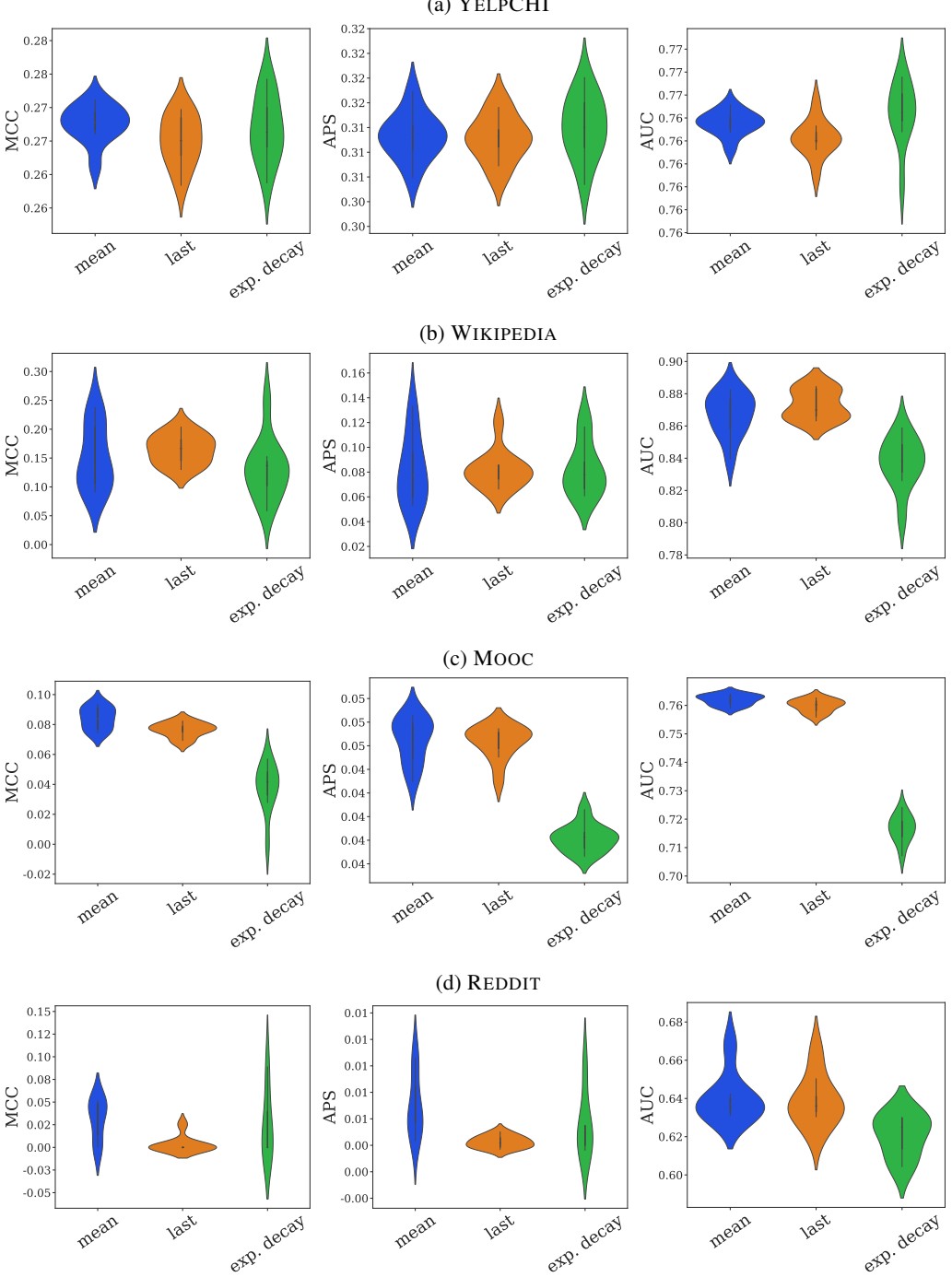

Figure 7: Aggragator Versions

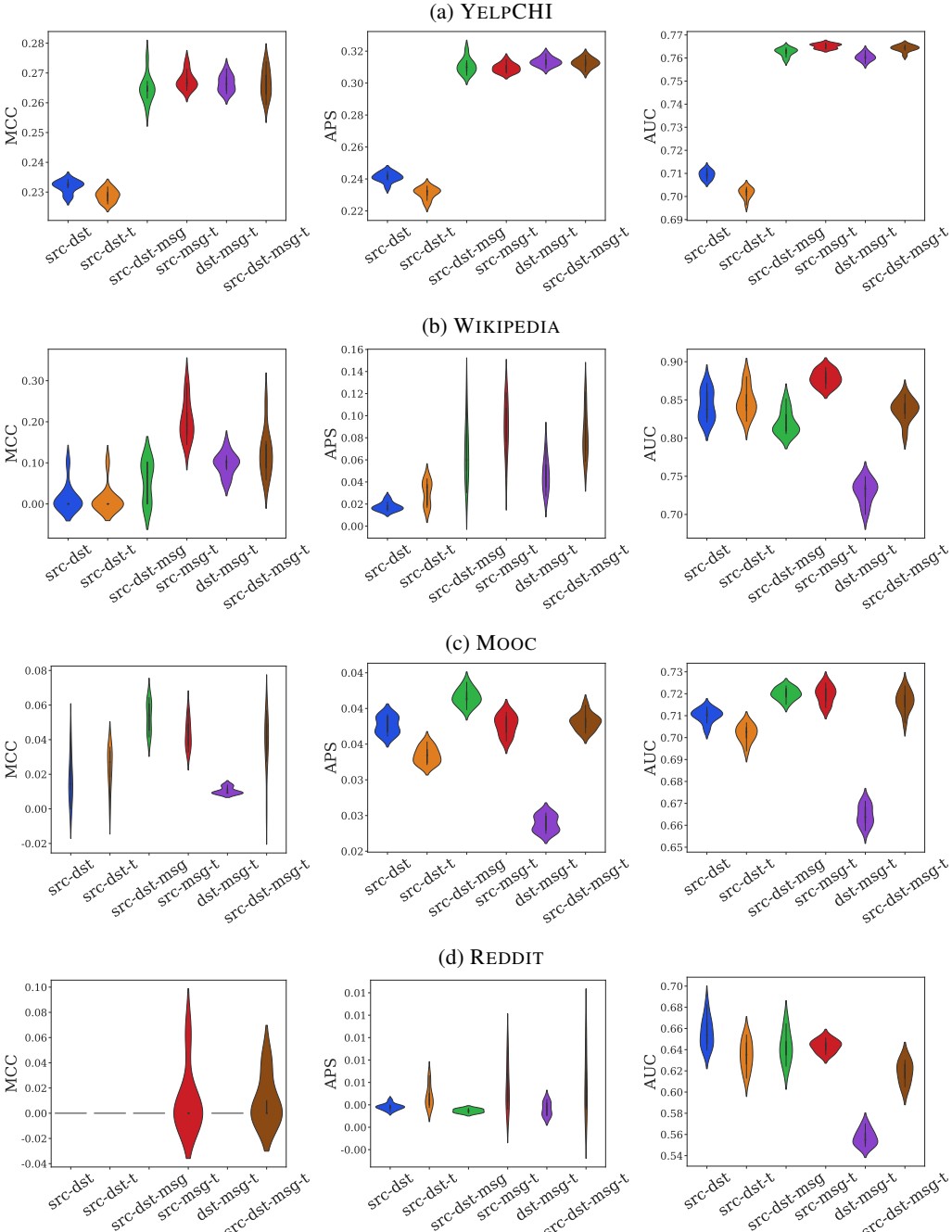

Figure 8: Readout Versions

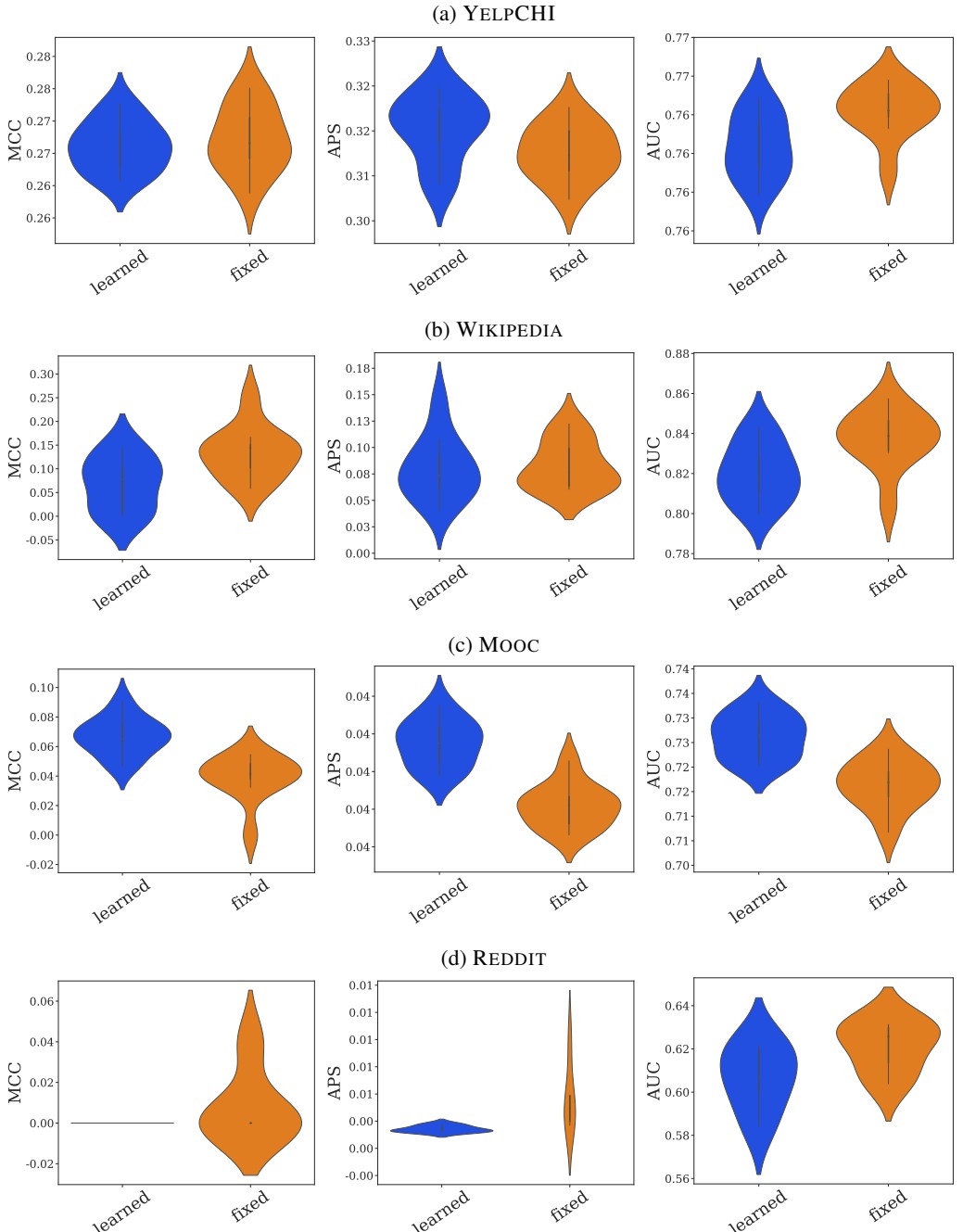

Figure 9: Time Encoding Versions

## F  READOUT VARIATIONS ON RLC

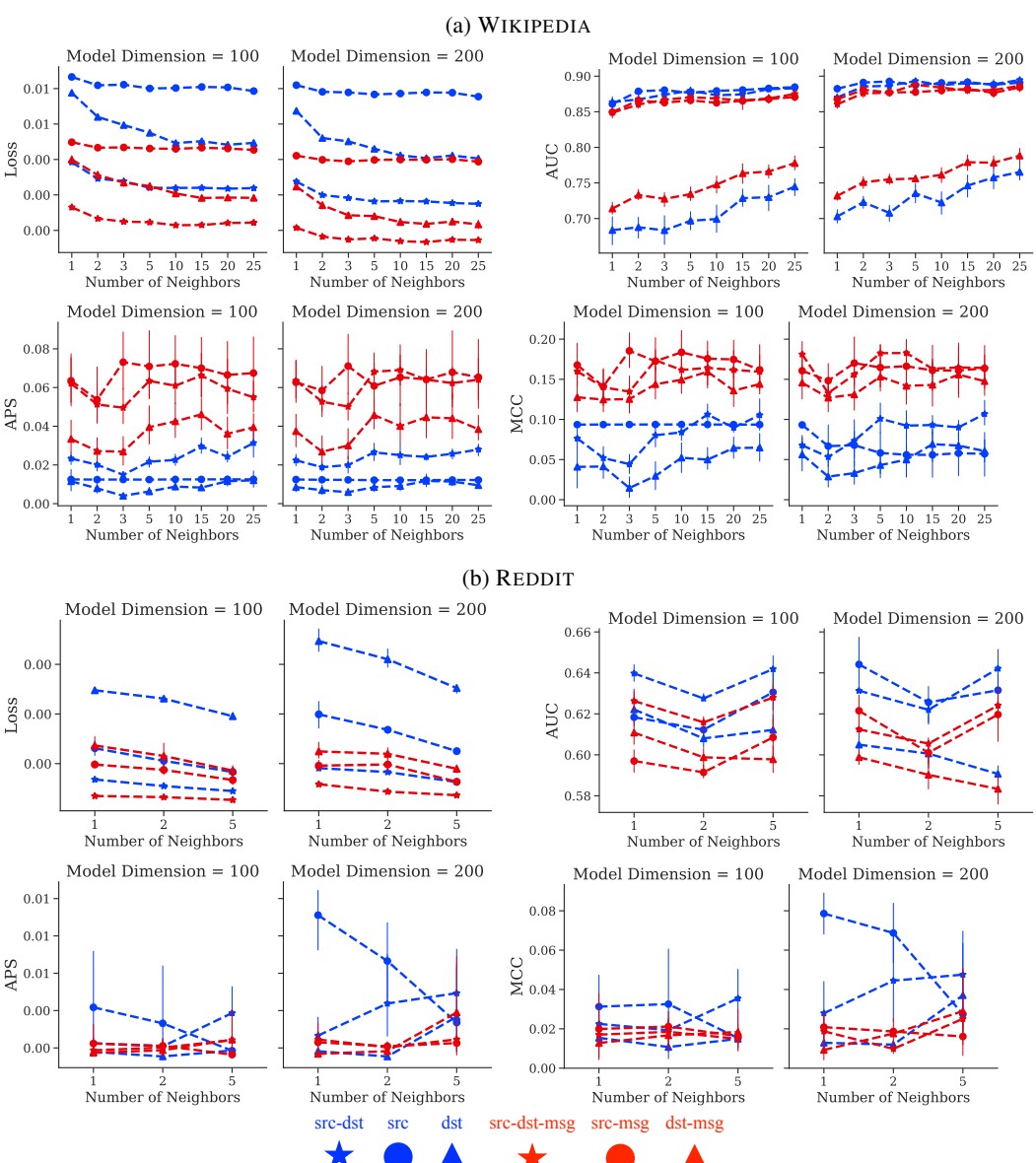

Figure 10: The performance on RLC using different variations of readout layer

# G    COMPARISON OF FLP AND RLC

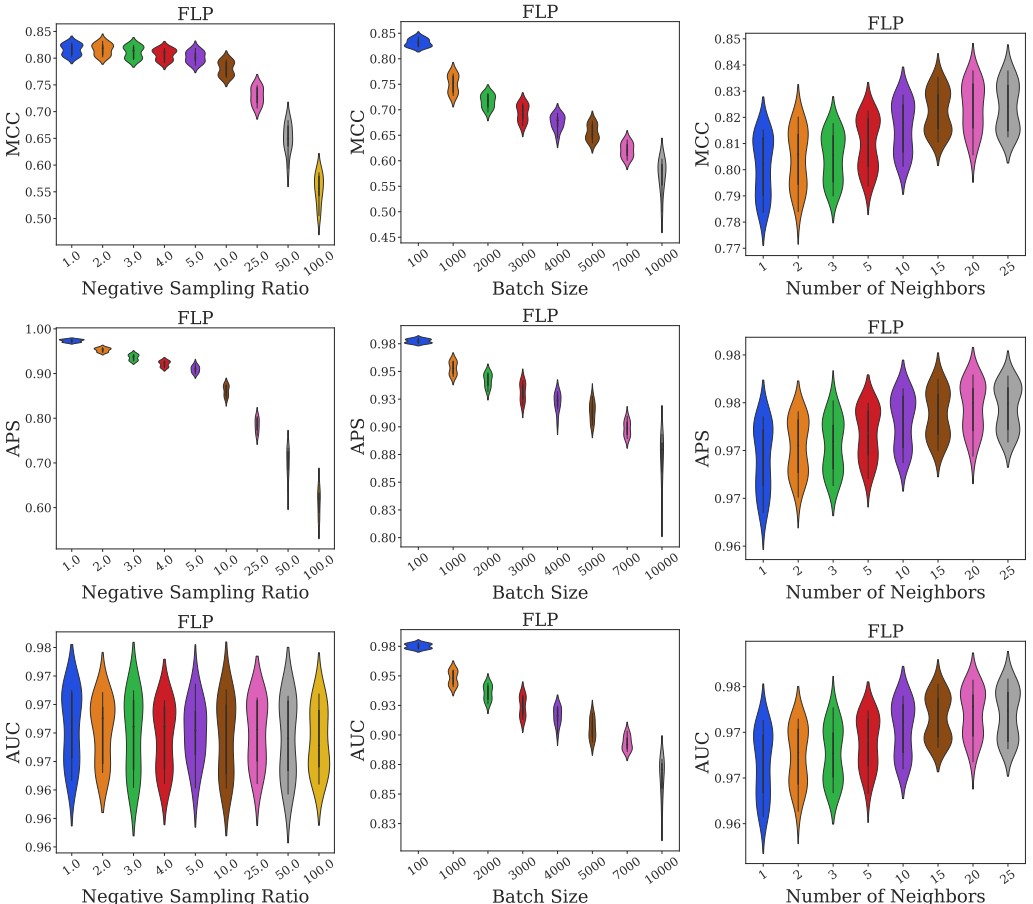

Figure 11: Parameter Sensitivity of FLP - WIKIPEDIA

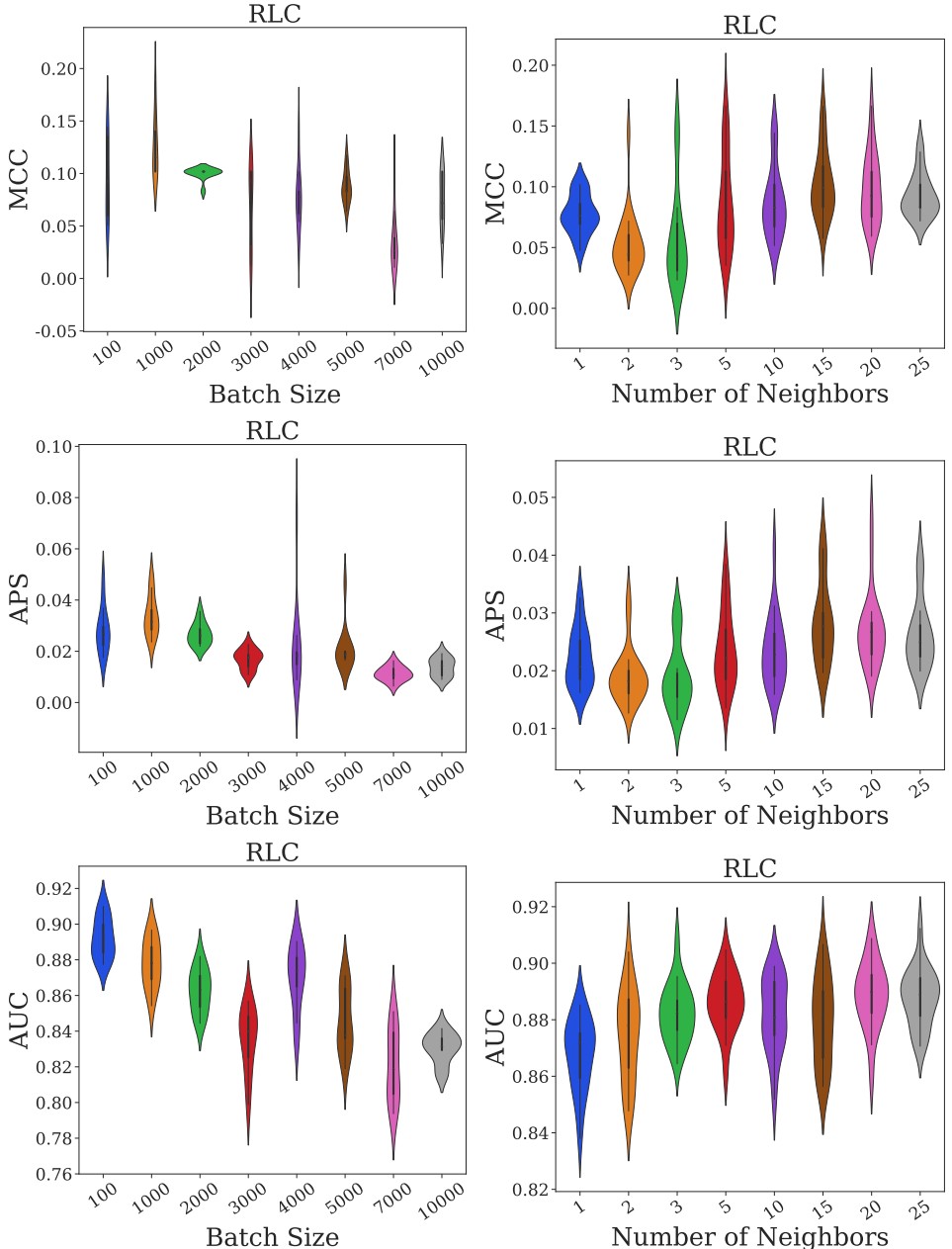

Figure 12: Parameter Sensitivity of RLC - WIKIPEDIA

