# OpenReview forum: "Recent Link Classification on Temporal Graphs Using Profile Builder"
_ICLR.cc/2024/Conference — Submitted to ICLR 2024_

### Official Review · Reviewer_HrQ1 · 2023-10-27

**Soundness:** 1 poor
**Presentation:** 1 poor
**Contribution:** 1 poor
**Rating:** 1
**Confidence:** 2

**Summary:**

In this paper, the authors introduced recent link classification (RLC), a new inference task on dynamic graphs in addition to temporal link prediction (TLP) and dynamic node classification (DNC), and formalized it as a benchmark downstream task. A new graph profiler method was then introduce to tackle RLC. Moreover, the authors also proposed new quality metrics (e.g., edge homophily and Matthews correlation coefficient). Experiments on a set of dynamic graph datasets preliminaries validated the quality of the proposed method w.r.t. the RLC task.

**Strengths:**

S1. The idea of treating recent link classification (RLC) as a new benchmark task of temporal graph learning seems interesting.

S2. The authors introduced new quality metrics (i.e., edge homophily and Matthews correlation coefficient).

S3. The authors provided the source code of their experiments.

**Weaknesses:**

**W1. The motivations of some statements and designs are unclear.**

In Section 1, why the fact that 'FLP is sensitive to non-architectural hyperparameters' can begs a question regarding DNC, i.e., 'in analogy to the dynamic node classification task, is there a temporal link classification task we can define?' From my perspective, the relationships between FLP and RLC are also not fully discussed. At the very beginning of the paper, it is highly recommended to add a toy running example (e.g., a simple dynamic graph) to illustrate what are FLP, DNC, and RLC, as well as their inherent relationships.

In the design of profile encoder, what are the motivations to introduce metapaths? As I known, metapaths are usually used in the graph representation learning of heterogeneous graphs, but it seems that the authors only consider (dynamic) homogeneous graphs in this study. Moreover, metapaths are also not illustrated in Fig. 1 (i.e., the model architecture).

There are no intuitive motivations regarding the proposed metrics of edge homophily and Matthews correlation coefficient (e.g., why Matthews correlation coefficient can handle the label-imbalanced issue). As a results, it is unclear what are their advantages beyond conventional quality metrics.

***

**W2. The problem statements in Section 3 are unclear and even confusing. Some presentation and statements seem to be inconsistent.**

In the 1st paragraph of Section 3, the availability of graph attributes (e.g., inputs of node and edge features) are not mentioned. However, as stated in the 2nd paragraph, edges attributes are treated as inputs of RLC. It is unclear that whether graph attributes (in terms of node attributes/features or edge attributes/features) are considered in this paper. If so, are they assumed to be static (for all time steps) or they are also dynamic?

At the very beginning of Section 3, it is suggested to highlight which data model (i.e., discrete-time dynamic graph or continuous-time dynamic graph) that the authors adopted in this paper.

The formal definitions of FLP and DNC are not given. For both FLP and DNC, there are transductive and inductive settings. The formal definitions regarding the transductive and inductive of RLC are not given. It is also unclear that the authors only consider the transductive setting or both transductive and inductive settings.

'Profile' is a significant concept in the proposed method, e.g., profile encoder, node profile, etc. However, there seems no definition regarding this concept (e.g., what are profiles in real dynamic graphs and in terms of what).

According to the statements in Section 3, each edge in a (dynamic) graphs should be associated with a time step. However, the graphs in Fig. 1 and Fig. 2 seem to be static. Furthermore, edge attributes are also not illustrated in Fig. 1 and Fig. 2.

***

**W3. Experiments are too simple. Some details regarding experiment settings are also unclear.**

In experiments, there are only two baseline methods (i.e., TGN and GraphMixer), which cannot fully validate the superiority of the proposed method. In addition to the two baselines, there are also some other dynamic graph representation learning methods (e.g., TGAT, DySAT, EvolveGCN, etc.) as mentioned in Section 2 that can be included in experiments. Experiment results of GraphMixer are not given in Table 3, Fig. 4, etc. In Table 1, the number of timesteps and the number of classed are not given for each dataset. The quality metric w.r.t. the results in Table 3 is not mentioned in the caption.

***

**W4. The major contributions of this paper are unclear and not fully verified.**

Although the authors claimed that they proposed a new temporal graph learning task (i.e., RLC) and new quality metrics (i.e., edge homophily and MCR), their advantages beyond existing techniques (e.g., what are the advantages of treating RLC as a new temporal graph learning task beyond FLP and DNC) are not fully discussed in the paper and not fully validated in experiments, due to the unclear motivations and insufficient experiments.

***

**W5. The overall presentation is poor. In addition to the inconsistent presentation mentioned before, there are also some grammatical errors and typos that need careful revisions.**

1) 'analyze the temporal graph learning architectures divindign categorizing the methods literature into two groups'

2) 'edges$\mathcal{E}$'

3) 'we construct derived graphs that'

4) 'connect a vertex acting that acts as a source to another that acts as a course through a shared destination vertex'

5) 'on abuse-like like datasets'

**Questions:**

See W1, W2, and W4.

---

> ### Author Response · Authors · 2023-11-11
> **Proposed plan for revision**
>
> Thank you for your thorough review, and your specific feedback. We are developing a plan to address the weaknesses that you have identified, and we would like to confirm with you that this plan would address your concerns.
>
> - W1: We will add a simple illustrative example indicating the differences between FLP, RLC, and DNC.
> - W1: The graphs that are currently considered are all bipartite and heterogeneous with interactions typically being between a user → item. This is not required for the RLC task in general, but is the structure of our dataset under consideration. In such a setting, we would argue that the metapath construction is a relatively natural fit.
> - W1: Metric uncertainty - We will add an illustrative example that highlights situations where MCC can correctly handle label imbalance but more traditional metrics cannot. In addition, we will leverage the edge homophily definition to explain aspects of our results, in analogy to the discussion of node homophily for node classification.
> - W2: We will rework the problem statement to address the lack of clarity. Specifically, we will make clear that we are working in a continuous time setting, the source of our edge features, and the transductive nature of the problem.
> - W2: We will clarify the definition of a profile in this setting, rather than depending on the colloquial definition.
> - W2: The graphs in the figures are illustrative but they are dynamic based on the content of the containers. We will make this more clear in the text and figure captions
> - W3: We will add two additional datasets, one based on video game reviews and one on NFT sales, and baselines from GraphMixer and TGAT. The NFT sales dataset is considerably larger than the ones in the current work.
> - W4: We will rework the text to highlight the importance of our new task and more clearly explain how it differs from FLP and DNC; and we will use that to further motivate our discussion about the value of this new task. We hope that by addressing the metrics as we have planned in W1, and clearly explaining the value of RLC, we can address W4.
> - W5: We thank the reviewer for pointing out typos and other places where the work lacks clarity. We will address those, and carefully edit the paper to catch others.
>
> We aim to have this work completed before the end of the review process, so that we can address any additional concerns that you may have. Please do let us know if this plan of action is insufficient to address your points.
>
> Best Regards,
>
> The Authors

---

> ### Author Response · Authors · 2023-11-18
> **Follow up & revision**
>
> We again thank the reviewer for their thorough review and specific feedback. We have just uploaded a new draft of the paper that has addressed the points that have been raised. Specifically we have:
> - W1: We have added such a figure as figure 1.
> - W1: We have added an illustrative example of a situation where MCC and APS give conflicting results, and provided an explicit citation to a paper that provides a clear argument for MCC over AUC.
> - W2: We have updated the problem definition provided in Section 3 in the light of your questions.
> - W3: We have added two additional datasets and two additional baselines in Table 3.
> - W4: We have made the difference between RLC, FLP, and DNC much more clear.
> - W5: We have given the paper a thorough edit, and we will perform one more pass of editing on Monday to catch any last typos.
>
> We welcome any further questions. Please don’t hesitate to share them so that we can address them as quickly as possible.
>
> Best Regards,
> The Authors

---

### Official Review · Reviewer_V529 · 2023-11-02

**Soundness:** 2 fair
**Presentation:** 2 fair
**Contribution:** 2 fair
**Rating:** 5
**Confidence:** 3

**Summary:**

This paper works on edge classification (Recent Link Classification) on dynamic graphs. It uses a metric, Matthews Correlation Coefficient for imbalanced datasets and benchmarks TGN (Rossi et al.,2020) on message aggregation schema, readout layer, and time encoding strategy. It then proposes Graph Profiler, which has better model performance than TGN.

**Strengths:**

1. Edge classification is an important topic.
2. Introduced critical design principles look helpful for algorithm design.
3. Experiments show Graph Profiler performs better than TGN.

**Weaknesses:**

1. The reason why Graph Profiler performs better than TGN is unclear. The technical advancement of Graph Profiler is unclear.
2. These critical design principles are different for different datasets, which makes it morel like hyper-parameter tuning for specific datasets.
3. Extensive evaluation (e.g. larger datasets, other models) are needed for validating these critical design principles.

**Questions:**

1. The motivation of the paper is unclear. If the authors want to highlight the proposed method, it might be better to explain how the design of the Graph Profiler algorithm incorporates these critical design principles.
2. What is the key takeaway for these critical design principles?
3. Do these critical design principles also fit other settings like node classification and future link prediction?

---

> ### Author Response · Authors · 2023-11-11
> **Proposed plan for revision**
>
> Thank you for your thorough review, and your specific feedback. We are developing a plan to address the weaknesses that you have identified, and we would like to confirm with you that this plan would address your concerns.
> - Our Graph Profiler method is novel in so far as, to our knowledge, the concept of building time-dependent profiles for entities based on based on the features and labels of their previous interactions  is not something we are aware of. In contrast, TGN learns a graph wide memory. We will update the text to make this point clear. Also, we believe that we have additional contributions, including the introduction of a new research task, an associated figure of merit, and baseline results. We will rework the text to further emphasize these contributions.
> - On the topic of extensive evaluations being required, we will add two additional datasets, one based on video game reviews and one on NFT sales, and baselines from GraphMixer and TGAT. The NFT sales dataset is considerably larger than the ones in the current work.
> - On the topic of critical design principles, we will rework the text to emphasize that these principles comprise the design space for a family of models.
> - We will include our investigations of the identified design space for the FLP in an appendix and make reference to it in the main text.
>
> We aim to have this work completed before the end of the review process, so that we can address any additional concerns that you may have. Please do let us know if this plan of action is insufficient to address your points.
>
> Best Regards,
>
> The Authors

---

> ### Author Response · Authors · 2023-11-18
> **Follow up & revision**
>
> We again thank the reviewer for their thorough review and specific feedback. We have just uploaded a new draft of the paper that has addressed the points that have been raised. Specifically we have:
> - We have cleaned up the text and notation, taking the time to be more clear about many of our definitions.
> - We have clarified the importance of the TGN modifications for motivating our design choices for our Graph Profiler model. This clarification can be found in the first paragraph of section 5.2. In summary, we explored modifications to TGN that borrowed specifics from other SOTA models, such as the fixed time encoder, to understand the importance of these model-features for our RLC task. Our development of Graph Profiler was informed by the important components of our modified TGN.
> - We have performed a related analysis to the above that explores the difference between RLC and FLP for a our modified TGN, and report this comparison in Appendix G.
> - We have included time dependent measures of our edge homophily in Appendix A. We hope that this provides a clearer perspective on the way that our datasets evolve with time.
> - We have added two additional datasets and two additional baselines in Table 3.
> - We have given the paper a thorough edit, and we will perform one more pass of editing on Monday to catch any last typos.
>
> We welcome any further questions. Please don’t hesitate to share them so that we can address them as quickly as possible.
>
> Best Regards,
> The Authors

---

### Official Review · Reviewer_Sc4M · 2023-11-05

**Soundness:** 2 fair
**Presentation:** 2 fair
**Contribution:** 2 fair
**Rating:** 5
**Confidence:** 3

**Summary:**

This paper formulates the problem of dynamic edge classification and proposes a method for this task and a specific metric for evaluating the performance of this task. Specifically, the proposed metric can handle the case of class imbalance, and the proposed model includes a novel message aggregation schema.

**Strengths:**

1. The problem of dynamic edge classification is important.

2. The problem formulation is coherent and well-reasoned.

3. The experiments conducted are thorough, with the authors exploring a wide range of variants.

**Weaknesses:**

1. The model design is rather conventional and lacks novelty. It adheres to the traditional message-passing framework and introduces event and time-related elements as a simple extension.

2. While the proposed Matthews Correlation Coefficient is effective for assessing classification tasks, it may not fully account for the specific attributes of the problem, particularly in the context of temporal interaction classification. It remains unclear how well it aligns with the temporal and graph-based nature of the problem.

3. It is recommended that the authors provide equations for all modules in the paper to offer a comprehensive understanding of the model. This would be especially beneficial in elucidating model details.

**Questions:**

1. Is there any novelty in the method design, such as in a specific module or the whole framework?

2. Is there any specific design of the metric for temporal interaction classification?

3. What is the time encoder like?

Please elaborate on the above issues to ensure that I don't miss any contributions in the paper.

**Details Of Ethics Concerns:**

No ethical issues were found.

---

> ### Author Response · Authors · 2023-11-11
> **Proposed plan for revision**
>
> Thank you for your thorough review, and your specific feedback. We are developing a plan to address the weaknesses that you have identified, and we would like to confirm with you that this plan would address your concerns.
> - Our Graph Profiler method is novel in so far as, to our knowledge, the concept of building time-dependent profiles for entities based on based on the features and labels of their previous interactions  is not something we are aware of. Also, we believe that we have additional contributions, including the introduction of a new research task, an associated figure of merit, and baseline results. We will rework the text to further emphasize these contributions.
> - We will address the typos through a thorough editing process.
> - We will add an illustrative example of a situation where MCC and AUC give conflicting results and provide a clear argument for MCC.
> - We will add more information about the time-encoder, and more generally we thank the reviewer for pointing this areas where the methods discussion is unclear, and we will work to improve that.
> - We will add 2 additional datasets and additional benchmark results to make the analysis a bit more involved.
>
> Additionally, we would appreciate if you could expand a little further on your question: “Is there any specific design of the metric for temporal interaction classification?” We are planning to answer it by expanding the MCC definition to explain that it applies in multi-class and imbalanced settings, but we are uncertain if this is a satisfactory answer for your question
>
> We aim to have this work completed before the end of the review process, so that we can address any additional concerns that you may have. Please do let us know if this plan of action is insufficient to address your points.
>
> Best Regards,
>
> The Authors

---

> ### Author Response · Authors · 2023-11-18
> **Follow up & revision**
>
> We again thank the reviewer for their thorough review and specific feedback. We have just uploaded a new draft of the paper that has addressed the points that have been raised. Specifically we have:
> - We have cleaned up the text and notation, taking the time to be more clear about many of our definitions.
> - We have added an illustrative example of a situation where MCC and APS give conflicting results, and provided an explicit citation to a paper that provides a clear argument for MCC over AUC.
> - We have added more information about the time encoder.
> - We have included time dependent measures of our edge homophily in Appendix A. We hope that this provides a clearer perspective on the way that our datasets evolve with time.
> - We have added two additional datasets and two additional baselines in Table 3.
> - We have given the paper a thorough edit, and we will perform one more pass of editing on Monday to catch any last typos.
>
> We welcome any further questions. Please don’t hesitate to share them so that we can address them as quickly as possible.
>
> Best Regards,
> The Authors

---

### Official Review · Reviewer_aigg · 2023-11-08

**Soundness:** 3 good
**Presentation:** 2 fair
**Contribution:** 2 fair
**Rating:** 5
**Confidence:** 4

**Summary:**

The paper proposes a new task, called Recent Link prediction, which calls for classifying a graph link that has already occurred. This comes in the context of various industrial applications such as predicting whether a transaction (user node interacting (via an edge) with a credit card node) is fraudulent. The authors formulate the learning task, propose an architecture, an evaluation metric and conduct several experiments.

**Strengths:**

Pros
* For the most part the paper is well written and provides intuition on the new setting proposed by the authors

* The proposal is backed by several experiments

* The idea is interesting, the contrast with current methods is discussed and the need for the new task is well justified.

**Weaknesses:**

Cons
* The technical details on the graph profiler are hard to follow, some notation is missing or assumed. At the same time the authors explain more basic concepts such as edge homophily using more verbage.

* The evaluation (since it is a novel task) is a bit weak. However, the analysis can be supplemented with more creative ways of analyzing the performance even if comparison to other algorithms is not possible.

**Questions:**

* Given the datasets/tasks you are describing, it appears that these graphs are knowledge graphs (consisting of entities and relations connecting them). If my understanding is correct, how does this new task relate to the dynamic knowledge graph link prediction?

* The edge homophily paragraph is dense with notation, which makes the formula hard to digest even though the idea is simple. Please include a sentence (in English) to supplement the formula when defining edge homophily, e.g. "the fraction of edges that connect nodes of the same
class".

*  Are there other metrics beyond edge homophily that are useful here? Since you don't take the time dimension in the edge homophily, does any other metric make sense for the evaluation of the time component?

* Please define the matrices you use (and their dimensions), they are sometimes only understood from the context, e.g. in the Profiler Encoder section on p.4

* How did you derive the formulat for d_1, bottom of p. 4?

* Some typos need to be fixed: e.g.

     * p.2 "...temporal graph learning architectures divinding categorizing..."
     * p.4 "...In our specific instance... acting that acts..."

* What is the significance of the TGN modifications in Table 2? They don't seem to be directly related to the proposed method.

---

> ### Author Response · Authors · 2023-11-11
> **Proposed plan for revision**
>
> Thank you for your thorough review, and your specific feedback. We are developing a plan to address the weaknesses that you have identified, and we would like to confirm with you that this plan would address your concerns.
> - We thank the reviewer for pointing this areas where the methods discussion is unclear, and we will work to improve that. We will do so specifically by expanding upon details of our model, dimensions of matrices, and other model details; as well as the way in which we’ve constructed the dataset.
> - We will expand our definition of edge homophily to be easier to understand.
> - We will add 2 additional datasets and additional benchmark results to make the analysis a bit more involved.
> - Knowledge Graph Link Prediction (KGLP) is a related problem but is different for multiple reasons. The first is that in RLC, the prediction is performed conditioned on the existence of the link. The second is that KGLP implies an ontological structure to the relationships which can naturally be leveraged.
> - As for the question of additional metrics, we will add plots of the edge-homophily over time and a metric that we are referring to as balanced edge homophily.
> - We will address the typos through a thorough editing process.
> - We will review the rest of the paper to ensure that formulas like that for $d_1$ are motivated and clear.
>
> We aim to have this work completed before the end of the review process, so that we can address any additional concerns that you may have. Please do let us know if this plan of action is insufficient to address your points.
>
> Best Regards,
>
> The Authors

---

> ### Author Response · Authors · 2023-11-18
> **Follow up & revision**
>
> We again thank the reviewer for their thorough review and specific feedback. We have just uploaded a new draft of the paper that has addressed the points that have been raised. Specifically we have:
> - We have cleaned up the text and notation, taking the time to be more clear about many of our definitions.
> - We have clarified the definition of edge homophily by adding a paragraph that discusses its intuitive definition.
> - We have included time dependent measures of our edge homophily in Appendix A. We hope that this provides a clearer perspective on the way that our datasets evolve with time.
> - We have added two additional datasets and two additional baselines in Table 3.
> - We have given the paper a thorough edit, and we will perform one more pass of editing on Monday to catch any last typos.
> - We have clarified the importance of the TGN modifications for motivating our design choices for our Graph Profiler model. This clarification can be found in the first paragraph of section 5.2.
> - We thank the reviewer for directing us to the link between RLC and Dynamic KG link prediction These tasks are quite related, but our current formulation of RLC has the additional complexity of delayed label observations. We have formulated this task in this way to match common industrial tasks such as fraud or toxicity detection.
>
> We welcome any further questions. Please don’t hesitate to share them so that we can address them as quickly as possible.
>
> Best Regards,
> The Authors

---

> > ### Comment · Reviewer_aigg · 2023-12-04
> > **response to comments**
> >
> > Dear authors, thank you very much for all the comments and the work put towards all the modifications. While your comments address my questions, I will keep an eye on the other reviewers responses as I think they are brining in very valid points and concerns.

---

### Official Review · Reviewer_cW29 · 2023-11-10

**Soundness:** 3 good
**Presentation:** 3 good
**Contribution:** 2 fair
**Rating:** 5
**Confidence:** 3

**Summary:**

The paper introduces the "Recent Link Classification" task within the field of "Temporal Graph Learning," focusing on categorizing existing links based on source and destination entities. The authors evaluate baseline methods for future link prediction and temporal graph learning in recent link classification, employing the Mathews Correlation Coefficient as the evaluation metric. Their proposed Graph Profiler architecture consists of five components: profile encoder, message encoder, destination encoder, and a readout layer for information aggregation. The study delves into various strategies for the profile encoder, time encoder, and readout layer, demonstrating performance enhancements over baseline methods.

**Strengths:**

1. The proposed recent link classification task is practical and holds applicability for addressing real-world industrial problems.
2. The paper conducts a comprehensive investigation into the combination of different approaches from temporal graph learning literature.

**Weaknesses:**

1. The modeling decisions, such as the choice between learnable or fixed time encoding, appear ad-hoc and contingent on specific datasets. It would be beneficial to elucidate insights or provide general guidance for determining an optimal combination on new datasets. For instance, what factors contribute to the observed performance variation, and is there a rationale for the less effective performance of learnable time-encoding on the Wikipedia dataset?
2. Given that the src-dst-msg-t configuration doesn't generally yield the best results, I am wondering about the necessity of introducing seemingly redundant components like the destination encoder. Additionally, the observed performance degradation in cases where time encoding is added to src-dst raises questions. Is there any explanation for that?
3. The presented task bears similarities to entity relationship classification in NLP. It would be interesting to discuss the similarities and differences between these two tasks.
4. The proposed method mainly combines several existing methods together, which makes the technical contribution of method design not very high.


Minor comment:
1. The paper contains several typos, and certain sentences are challenging to comprehend.
2. The captions in the graph are too small to read.

**Questions:**

Please refer to the weaknesses.

---

> ### Author Response · Authors · 2023-11-11
> **Proposed plan for revision**
>
> Thank you for your thorough review, and your specific feedback. We are developing a plan to address the weaknesses that you have identified, and we would like to confirm with you that this plan would address your concerns.
> - We will work to understand the structure of the datasets under investigation and how their structure impacts the time encoding.
> - To tease out the importance of the destination encoder, we will perform an ablation study and report those results. In addition to the destination encoder ablation study, we will perform other ablation studies and results in the main text of the paper.
> - We will explore the link between RLC and dynamic KG relationship classification. We thank the reviewer for pointing this connection out as the authors are unfamiliar with this literature.
> - Our Graph Profiler method is novel in so far as, to our knowledge, the concept of building time-dependent profiles for entities based on the features and labels of their previous interactions is not something we are aware of. Also, we believe that we have additional contributions, including the introduction of a new research task, an associated figure of merit, and baseline results. We will rework the text to further emphasize these contributions.
> - To address the figure captions, we will remake the figures such that they are easier to read.
> - We will address the typos through a thorough editing process.
> - We will add 2 additional datasets and additional benchmark results.
>
> We aim to have this work completed before the end of the review process, so that we can address any additional concerns that you may have. Please do let us know if this plan of action is insufficient to address your points.
>
> Best Regards,
>
> The Authors

---

> ### Author Response · Authors · 2023-11-18
> **Follow up & revision**
>
> We again thank the reviewer for their thorough review and specific feedback. We have just uploaded a new draft of the paper that has addressed the points that have been raised. Specifically we have:
> - Performed an ablation study to understand the role of destination embeddings, which can be found in Appendix C. We observe that in 5/6 datasets, the destination embeddings help, and we present an explanation for why this would be true.
> - We have explored the structure of the datasets and unfortunately have no explanation for the importance of the time encodings in the TGN framework. We do want to make clear, however, that the Graph Profiler model only ever uses learnable time encodings.
> - We have added two additional datasets and two additional baselines in Table 3.
> - We have given the paper a thorough edit, and we will perform one more pass of editing on Monday to catch any last typos.
> - We thank the reviewer for directing us to the link between RLC and Dynamic KG relationship classification. These tasks are quite related, but our current formulation of RLC has the additional complexity of delayed label observations. We have formulated this task in this way to match common industrial tasks such as fraud or toxicity detection.
>
> We welcome any further questions. Please don’t hesitate to share them so that we can address them as quickly as possible.
>
> Best Regards,
> The Authors

---

### Author Response · Authors · 2023-11-21
**General Response**

Dear Reviewers,

In general, thank you very much for your thorough reviews and feedbacks. In the light of your comments, we have revised our paper. The main changes are (1) experimentation on two new datasets that are processed specifically for the proposed benchmark task; Recent Link Classification (RLC) on temporal graphs, (2) adding more baseline comparisons, and (3) clarifying context of each experimental setting, in addition to minor fixes on notation and text, as you have suggested. As the rebuttal period concludes, we believe that we have effectively addressed all your inquiries and questions. Should you have any additional questions or require further clarification, please feel free to reach out. We appreciate your time and consideration and we would appreciate if you could revise your scores based on the changes made.

Best regards,

The Authors

---

### Meta-Review · Area_Chair_31js · 2023-12-06

**Metareview:**

I recommend to reject this paper.

  In this paper, the authors focused on the "recent link classification" task, which is an under explored task within the area of temporal graph learning. The authors proposed Graph Profiler architecture which consists of several components to encode profile (source node), message (feature vector of an edge), and destination node, and a readout layer for information aggregation. In addition to the traditional AUC and APS, the authors also urged to use Matthews Correlation Coefficient (MCC) as the   evaluation metric. The authors also conducted experiments to compare Graph Profiler with some of the STOA methods developed for other tasks in temporal graph learning (such as TGN and GraphMixer).

  All the reviewers recognized the importance of the task of recent link classification in the field of the temporal graph learning. However, they also identified a few weakness of this paper: 1) limited technical contribution/novelty in the design of GraphProfiler (Reviewer Sc4M, Reviewer cW29, Reviewer V529, & Reviewer HrQ1), 2) the weak evaluation to validate the critical design principles proposed in this paper, and 3) it is not clear to see the connection MCC in the RLC task. As a result, after the rebuttal, all the reviewers still think that this paper is below the acceptance threshold.

**Justification For Why Not Higher Score:**

I have read the paper and agreed with the reviewers that this paper is below the acceptance threshold due to the lack of enough technical contribution.

**Justification For Why Not Lower Score:**

N/A

---

### Decision · Program_Chairs · 2024-01-16

Reject